

# Tempo-spatial distribution of nitrogen dioxide within and around a large-scale wind farm-a numerical case study

**Jingyue Mo**[1,2]**, Tao Huang**[1]**, Xiaodong Zhang**[1]**, Yuan Zhao**[1]**, Xiao Liu**[2]**,**
**Jixiang Li**[1,2]**, Hong Gao**[1]**, Jianmin Ma**[1,3,4]**, Zuohao Cao**[5]

[1]Key Laboratory for Environmental Pollution Prediction and Control, Gansu Province, College of Earth and Environmental Sciences, Lanzhou University, Lanzhou, China
[2]College of Atmospheric Sciences, Lanzhou University, Lanzhou, China
[3]Laboratory for Earth Surface Processes, College of Urban and Environmental Sciences, Peking University, Beijing, China
[4]CAS Center for Excellence in Tibetan Plateau Earth Sciences, Chinese Academy of Sciences, Beijing, China
[5]Weather Forecasting Research Division, Environmental and Climate Change Canada, Toronto, Canada

*Correspondence to:* Jianmin Ma (jianminma@lzu.edu.cn)

**Abstract.** As a renewable and clean energy, wind power has become the most rapidly growing energy resource worldwide in the past decades. Wind power has been thought not to exert any negative impacts on the environment. However, since a wind farm can alter the local meteorological conditions and increase the surface roughness lengths, it may affect air pollutants passing through and over the wind farm after released from their sources and delivered to the wind farm. In the present study, we simulated the nitrogen dioxide ($NO_2$) air concentration within and around a world's largest wind farm (Jiuquan wind farm in Gansu Province, China) using a coupled meteorology and atmospheric chemistry model WRF-Chem. The results revealed an "edge effect", which was featured by higher $NO_2$ levels at the immediate upwind and border region of the wind farm and lower $NO_2$ concentration within the wind farm and the immediate downwind transition area of the wind farm. A surface roughness length scheme and a wind turbine drag force scheme were employed to parameterize the wind farm in this model investigation. Modeling results show that the both parameterization schemes yield higher concentration up to 34% in the immediate upstream of the wind farm and lower concentration within the wind farm compared to the case without the wind farm. We infer this edge effect and the spatial distribution of air pollutants to be a result of the internal boundary layer induced by the changes in wind speed and turbulence intensity driven by the rotation of the wind turbine rotor blades and the enhancement of surface roughness length over the wind farm. The step change in the roughness length from the smooth to rough surfaces (overshooting) in the upstream of the wind farm decelerates the atmospheric transport



of air pollutants, leading to their accumulation. The rough to the smooth surface (undershooting) in the downstream of the wind farm accelerates the atmospheric transport of air pollutants, resulting in lower concentration level.

## 1  Introduction

Wind power has been the fastest-growing energy and one of the most rapidly expanding industries in the globe. To fulfill the sustainable development, establish an "environment-friendly society" and reduce emissions of $CO_2$ and other air pollutants, considerable efforts have been made in China to develop and expand wind power generation in the past decade. China's wind power has increased 100% from 2006 to 2010. By 2015, the total installed capacity of wind power has become the largest globally with the capacity of 140GW (GWEC, 2016). It is projected that wind power capacity in the nation will reach 200 GW by 2020, 400 GW by 2030, and 1000 GW by 2050. In 2016, the wind power capacity has accounted for 4% of total national electricity consumption. It is expected that wind power will become one of five main power sources and meet 17% of the total electricity demand in China in the mid-21st century (IEA, 2013).

Extensive field and modeling studies have demonstrated that a relatively large-scale wind farm could alter the local meteorological and climate conditions. From a dynamic perspective, a large-scale wind farm can be approximately regarded as a sink of kinetic energy (KE) and source of turbulent kinetic energy (TKE). Turbulence generated by wind turbine rotors could create eddies that can enhance vertical mixing of momentum, reducing the wind speed at the turbine hub-height level (Baidya et al., 2004; Baidya, 2011; Barrie et al., 2011). The wind farm induced turbulence can also alter the vertical mixing which can markedly affect the vertical distribution of temperature and humidity (Baidya et al., 2004; Baidya, 2011). Coupled atmosphere-ocean climate model has predicted that the global distribution of wind farms could enhance air temperature by up to 1℃ in inland wind farms and cool down temperature near the ground surface by 1℃ in offshore wind farms (Keith et al., 2004). Ocean-atmosphere heat fluxes would increase to respond increasing turbulence produced by wind farms (Barrie et al., 2010). Nevertheless, although the effects of wind farms on meteorology have been observed and simulated, overall the net impact of wind power on global surface temperatures could be overlooked (Wang and Prinn, 2010). Satellite remote sensing and model simulations confirmed that the degree of variations in the surface temperature altered by large-scale wind farms were not significant compared to the benefit from wind power in the emission reduction of $CO_2$ and other greenhouse gases (Barrie et al., 2010; Keith et al., 2004; Baidya et al., 2010; Zhou et al., 2012).

As a clean energy, a wind farm does not release any harmful chemicals into the air and hence has not been paid particular attentions in the scientific community compared to its negative environmental



impacts on the wildlife, noise, visual impact (Saidur et al., 2011; Colby et al., 2009; Magoha, 2002; Loss et al., 2013), and meteorological and climate conditions. Wind farms could alter the underlying surface characteristics, and disturb winds and turbulence near and within the wind farms by enhancing the surface roughness length through wind turbines set up and spinning wind turbine rotors. These changes mostly occur near the surface or the atmospheric boundary-layer where the levels of air pollutants are highest. As a result, the wind power operation might affect the atmospheric transport and diffusion of an air pollutant released from its industrial and mobile sources near the wind farm. Further, considering the fast expansion of wind energy industry in the past and future, a question may arise: does the increasing number of wind farms would likely perturb local, regional, and national air pollution forecasting?

The effect of the wind farm on air pollution depends on several factors, including the source locations, proximity and strength, wind speed and direction, wind turbine size and the layout of wind turbines in the wind farm. It is not straightforward to measure the perturbations of an air pollutant induced by a wind farm. As an alternative, the present study made use of a coupled weather forecast and atmospheric chemistry model to simulate the air pollution within and around a large-scale wind farm subject to a typical atmospheric transport event of air pollutants emitted from a point source near the wind farm, aiming to 1. assess and quantify the temporal evolution and spatial distribution of the air pollutant within and around the wind farm; 2. evaluate the wind and turbulent fields that drive the spatial-temporal variation of the air pollutant over the wind farm, and 3. identify primary characteristics of the air pollutant in the wind farm under a specific mesoscale circulation over the wind farm. Results are reported below.

## 2. Materials and Methods

### 2.1 Locations of wind farm and major emission source

The location of the selected wind farm in this study is illustrated in Fig.1. This wind farm extends from Yumen (40°16' N, 97°02' E) to Guazhou (40°31' N, 95°42' E) in Jiuquan, located in the west end of Hexi Corridor, Gansu Province, northwestern China (Fig. 1a). Given its huge wind energy resources, Jiuquan region has been termed "The Land Three Gorges" (The Three Gorges is the largest hydroelectric power station in the world). The Jiuquan wind farm, which consists of Yumen wind farm (YWF) and Guazhou wind farm (GWF), has been ranked as the largest wind farm in the world (Fig. 1b).The total cumulative wind power energy was about 12 GW by 2015 and is projected to reach 13.6 GW by 2020. The wind turbine hub height in the YWF and GWF ranges from 70－90 m and rotor diameter ranges from 83 to 113 m (CCER, 2016), respectively. This large-scale wind farm covers an



area about 2000 km$^2$. The underlying surfaces over the YWF and GWF are almost entirely covered by the Gobi desert and bare lands with only several pieces of lands by residential area. The terrain height in the wind farm ranges from 1.2km to 2km above the sea level. The both YWF and GWF are located closely in the suburb of Jiuquan and Jiayuguan, the two largest cities in the Hexi Corridor. The largest
emission source of air pollutants proximate to the Jiuquan wind farm (YWF and GWF) is the Jiuquan Iron & Steel Group Co., Ltd (JISCO), located in Jiayuguan City (39°48' N, 98°18' E), about 110 km southeast to the YWF (Fig. 1b). This company is ranked as the largest Iron & Steel industry in northwestern China and one of the top 50 iron and steel companies in the world.

## 2.2    WRF-Chem Model setup and configuration

We applied WRF-Chem model v3.7 (http://www2.mmm.ucar.edu/wrf/users/wrfv3.7/wrf_model.htm) to simulate the meteorological field and atmospheric chemistry. The WRF-Chem (the Weather Research and Forecasting model coupled with Chemistry) is a new generation air quality model with its air
quality component (Chem) and meteorological component (WRF) being fully coupled in an "online" approach (Peckham et al., 2011). The physical options in WRF-Chem v3.7 include the Lin microphysics scheme (Lin et al., 1983), the Rapid Radiative Transfer Model (RRTM) longwave radiation scheme (Mlawer et al., 1997), Goddard shortwave scheme (Kim and Wang, 2011), revised MM5 M-O surface layer scheme (Beljaars,1994; Chen and Dudhia, 2001), YSU (Yonsei University)
boundary layers scheme (Hong et al., 2006), new Grell cumulus scheme (Grell and Devenyi, 2002), and Unified Noah land surface model (Chen and Dudhia, 2001). The chemical options include Madronich TUV, F-TUV, and Fast-J (Fast et al., 2005) photolysis scheme, modified CB05 gas-phase chemistry scheme with updated chlorine chemistry (Yarwoodetal., 2005), several photo chemical mechanisms by RADM2 (Middleton et al., 1990), CBMZ, and SAPRC, MEGAN biogenic emission scheme (Guenther
et al., 2012), and three aerosol modules, MADE/SORGAM, MOSAIC and a simple aerosol module from GOCART.

We used the anthropogenic emissions from HTAP_V2 (The Task Force on Hemispheric Transport of Air Pollution). This emission inventory consists of the gridded emission data and gridmaps of CH4, CO, SO2, NOx, NMVOC, NH3, PM10, PM2.5, BC and OC on 0.1° latitude × 0.1° longitude resolution.
The global gridmaps are a joint effort from the USEPA, the MICS-Asia group, EMEP/TNO, as well as the REAS and the EDGAR group. The bio-emission calculated by MEGAN V2.1 has a spatial resolution of 1 km (Model of Emissions of Gases and Aerosols from Nature). The FNL reanalysis data with 0.25° × 0.25° lat/lon provided by the National Centers for Environmental Prediction/National Center for Atmospheric Research (NCEP/NCAR) were used as initial and lateral boundary conditions.
Three nested domains on 10 km, 3.3 km, and 1.1 km resolutions were set up. The first domain (d01)





with 10 km spacing and an area of 850 km × 750 km covers Gansu Province and the part of Xinjiang Province. The second domain (d02) with 3.3 km spacing and an area of 413 km × 253 km covers Guazhou and Yumen wind farm. The third domain (d03) with 1.1 km spacing and an area of 124 km × 124 km covers Yumen wind farm only. The spatial configurations of these three model domains are

illustrated in Fig. 2. The fine domain lateral boundary conditions for the meteorological variables and air pollutants are interpolated from the coarse domain prediction. Two-way nesting is then optionally achieved by having the finegrid solution replace the coarse grid solution for those grid nodes that lie within the fine nest domain. The model has 28 eta levels with the top of 100 hpa. The vertical resolutions are much denser near the surface with 13 eta levels in the lowest 1km of the model

atmosphere (about 10m, 40m, 75m, 100m, 130m, etc.) so as to achieve more accurate simulations of meteorology and atmospheric chemistry in the planetary boundary layer (PBL).

## 2.3   Wind farm parameterization scheme

Two wind farm parameterization schemes were adopted to parameterize winds and turbulence fields forced by the wind turbines across the wind farm. The first one is the surface roughness length parameterization. In this scheme, a wind farm can be seen to increase underlying surface obstacles which reduce the wind speed in the hub height, featured by the increase of the aerodynamics roughness length (Baiyda et al., 2004; Keith et al., 2004; Oerlemans et al., 2007). Some of the previous model

studies were conducted by increasing the surface roughness lengths to quantify the aerodynamic effect of wind turbines on wind and turbulence profiles (Frandsen, 1992; Baidya et al., 2004; Keith et al., 2004). We adopted a similar approach to enhancing the roughness lengths over the GWF and YWF. To do so, we modified the geo-data in the WPS and the LANDUSEF table in WRF-Chem model. The roughness lengths in the wind farm were calculated using the Lettau roughness length equation (Lettau,

25   1965):

$$z_0 = 0.5h^* \frac{s_S}{s_L} \tag{1}$$

where $z_0$ is the roughness length in meters, $h^*$ is the average vertical extent of the roughness elements or

effective obstacle height (m). In our case, $h^*$ is the height of the wind turbine rotor. $S_S$ in Eq. (1) is the average silhouette area (m$^2$) of the average obstacle or the vertical cross-section area presented to the wind by one wind turbine; and $S_L$ is the density of roughness element. Here $S_L$ can be expressed as $S_L$= $A / N$, where $A$ is the area of the wind farm, and $N$ is the number of wind turbines (Porté-Agel et al., 2014; Rooijmans, 2004; Frandsen, 2007). For YWF, $h^*$ is taken as 113 m (wind turbine height), $S_S$ is



taken as 10,029 m$^2$, and $S_L$ is taken as 375,000 m$^2$. The resulted $z_0$ is 1.51m. We shall use this value as a typical roughness length to represent the underlying surface characteristics for the YWF. Knowing that bare land and Gobi desert are dominant underlying surface of YWF and its surrounding region, the roughness length on this surface was taken as 0.01 m outside the wind farm in model scenario simulations except for the control model run in which this surface roughness length was applied in entire d03 model domain (see below).

The second wind farm parameterization is the wind turbine drag force scheme, developed by Fitch et al. (2012) which was extended from Blahak et al. (2010) in the modeling of the conversion of KE from atmosphere wind flow (Fitch et al., 2012; Blahak et al. 2010). This scheme has been implemented in WRF model. The turbine drag force scheme was developed subject to the Mellor-Yamada-Nakanishi-Niino (MYNN) turbulence scheme (Mellor and Yamada, 1974; Nakanishi and Niino, 2009). The Fitch scheme takes into account the effects of the wind turbines on the atmospheric flow by adding a momentum sink on the wind flow and transferring the fraction of the KE from the atmosphere into electricity and TKE. The KE is quantified by a thrust coefficient $C_T$ which depends on the wind speed and the specification of the wind turbine. The electricity converted by KE is calculated by the power coefficient $C_P$ with change in the wind speed and varies between 17 – 75% of $C_T$. Both coefficients $C_T$ and $C_P$ can be obtained from a wind energy manufacturer. This approach assumes that the mechanical and electrical losses are negligible, so the KE could be transferred to TKE, given by $C_{TKE} = C_T - C_P$. The wind turbine drag force parameterization scheme reads

$$\boldsymbol{F}_{drag} = \frac{1}{2} C_T (|\boldsymbol{V}|)_\rho |\boldsymbol{V}| \, \boldsymbol{V} A, \tag{2}$$

$$\frac{\partial KE_{cell}^{ijk}}{\partial t} = \frac{\partial}{\partial t} \frac{\rho_{ijk} |V|_{ijk}^2}{2} (z_{k+1} - z_k) \Delta x \Delta y, \tag{3}$$

$$\frac{\partial P_{ijk}}{\partial t} = \frac{\frac{1}{2} N_t^{ij} C_P (|V|_{ijk}) |V|_{ijk}^3 A_{ijk}}{(z_{k+1} - z_k)}, \tag{4}$$

$$\frac{\partial TKE_{ijk}}{\partial t} = \frac{\frac{1}{2} N_t^{ij} C_{TKE} (|V|_{ijk}) |V|_{ijk}^3 A_{ijk}}{(z_{k+1} - z_k)}, \tag{5}$$

where $\mathbf{V} = (u, v)$ is the horizontal velocity vector, $\rho$ is the air density, $N_t$ is the density of wind turbines, A = $(\pi/4) D^2$ is the cross-sectional rotor area (where $D$ is the diameter of the turbine rotor), $i$, $j$, $k$ are number of grids in three-dimensional space (x, y, z), $\Delta x$ and $\Delta y$ are the horizontal grid spacing, and $z_k$ is the height of vertical coordinate. In the present study, the thrust coefficient $C_T = 0.16$, the turbine hub height is 90 m, and the rotor blade diameter is 113 m, nominal power of turbine is taken as 2.0 MW. These parameters are defined and implemented in the WRF files to parameterize the wind turbine



profiles.

To identify and quantify the influence of the YWF on air pollutants within and around this large-scale wind farm, we performed 4 model scenario runs. The first model scenario (S1) is the control run in which the YWF was not taken into consideration. Rather, we simply assigned the roughness length value of 0.01 m throughout the model fine domain (d03) including the YWF area. In the second model scenario (S2), the YWF was parameterized by the roughness length $Z_0 = 1.51$ m which was calculated by Eq. (1), and in the rest of the find model grids, $Z_0$ was taken as 0.01 m. In the third model scenario (S3), the YWF was parameterized by the drag force approach (Fitch et al., 2012) and the distance between two wind turbines is set to 500m. The last model scenario (S4) also made use of the drag force approach to parameterize the YWF, but the turbine density was extended from 500 m to 1 km.

## 2.4 A case study

From November 19th to 24th, 2016, a strong cold wave occurred in northern China. An anticyclone featured by a surface high pressure system moved from western Siberia to northern China. This system forced the change in the prevailing wind direction from westerly wind to easterly and southeasterly wind across the western Hexi Corridor on the south of the anticyclone. The air quality in Jiuquan city was deteriorated during this period, characterized by the rapid increase in atmospheric levels of several criteria air pollutants sampled at the Jiuquan air monitoring station which was operated by Ministry of Environmental Protection of China (http://www.zhb.gov.cn/). Given that both YWF and GWF are located in the northwest of Jiuquan City and Jiayuguan City (JISCO), heavy air pollutants from the JISCO were delivered to the two wind farms. We then performed extensive model investigations subject to the 4 model scenarios to assess numerically the tempo-spatial variation of air pollution in the YWF during this cold wave episode and heavy air pollution event. We selected $NO_2$ as the target air pollutant in the present investigation. While hourly sulfate dioxide ($SO_2$) concentrations were also available, its atmospheric level was lower than $NO_2$ due to the mandatory implementation of flue-gas desulfurization in JISCO, the major emission source of air pollutants in this region.

The modeling results from the model scenario 1 have been compared with the monitored $NO_2$ air concentrations from 0000 UTC November 19 to 0000 UTC November 21, 2016 at the Jiuquan Air Quality Monitoring Station operated by the Jiuquan Environmental Protection Agency. Figure S1 in Supplement shows the simulated and measured $NO_2$ concentrations at the Jiuquan Station. The statistics between the modeled and measured data are presented in Table S1. Overall, the model results agree reasonably well with the measured data but the modeled peak concentration lagged 4 hours behind the observed value.





## 3   Results

### 3.1   NO$_2$ in YWF without wind farm parameterization

Figure 3 shows simulated NO$_2$ air concentrations (ppmv) super imposed by the vector winds (m s$^{-1}$) at the first model vertical level (~10 m) across the fine domain (d03) at 0600, 1200, 2000 UTC November 19, and 0400 UTC November 20 from the model control run (model scenario 1, S1), respectively. At 0600 UTC (local time 1400), November 19, weak easterly winds prevailed over the most model domain, except in the south of the domain where northerly wind component prevailed (Fig. 3a). At this time,

NO$_2$ levels were low. At 1200 UTC, the southeasterly winds extending from the industrial source region (JISCO) to the YWF started to build up which delivered NO$_2$ from JISCO region to YWF (Fig. 3b). This southeasterly wind regime became stronger at 2000 UTC enhancing the atmospheric transport of NO$_2$ to the YWF, characterized by increasing NO$_2$ levels in the northwest of the JISCO and the YWF (Fig. 3c). The maximum NO$_2$ levels were observed in the wind farm between 2000 to 2300 UTC. Along

with the change in wind direction from southeast to northeast at 0400 UTC, November 20, NO$_2$ concentrations declined considerably compared to 2000 UTC, November 19 (Fig. 3d). Accordingly, Figure 4 illustrates the control scenario run predicted vertical cross section of hourly NO$_2$ concentrations from 1900 to 2200 UTC, November 19 along the transect across the fine domain (d03), highlighted by the red arrow line in Fig. 2. At 1900 UTC, the NO$_2$ plume extended from 0 to 25 km and moved from

southeast to northwest along the transect of YWF (Fig. 2). Relatively lower concentrations can be identified near the upwind interface of YWF (5-7 km, Fig. 4a), in line with of the pollutants moved towards the northwest. By next two hours at 2100 and 2200 UTC, the plume had moved to the upwind border of YWF (Fig. 4c, d), and remained there. The levels of NO$_2$ slightly increased from 1900 UTC (Fig. 4b, c). The results are in line with the horizontal advance of NO$_2$ concentrations near the surface,

as shown in Fig. 3c.

### 3.2   NO$_2$ in YWF due to roughness changes

Using the wind farm roughness length parameterization ($z_0 = 1.51$ m), we performed the second model

scenario run. Figure 5 shows the modeled hourly NO$_2$ concentrations at the same time as indicated in Fig. 3. Compared to the results from the control run, similar spatial patterns of NO$_2$ from the model scenarios 1 and 2 can be observed, characterized by northwest transport of NO$_2$ towards the YWF from its major industrial source to the southeast of YWF. However, the second model scenario run accounting for the roughness changes forced by the wind turbine setup appeared to yield higher NO$_2$ concentrations.

Considering that the atmospheric transport often dominates the spatial distribution of NO$_2$ under





prevailing winds, to identify the influence of the wind farm on $NO_2$ air concentrations, we simply estimated the concentration differences between the two model scenarios including and excluding the wind farm. Figure 6 illustrates the differences of $NO_2$ concentrations between the two model scenarios runs (the second model scenario run minus control run). As shown, the positive concentration

differences indicating higher concentrations from the second model scenario (S2) were found in the upwind and border region of the YWF and negative differences manifesting lower concentrations were identified within the YWF, particularly at 1200 and 2000 UTC. The mean positive concentration difference in the upwind region of the YWF is 0.009 ppmv. The estimated fraction $(C_{s2} - C_{s1}) / C_{s1} \times$ 100%, where $C_{s1}$ and $C_{s2}$ are mean concentrations from the model scenario 1 and 2 is 23%. The

negative concentration difference within the YWF is -0.009 ppmv and the ratio of the mean concentration from the second model scenario (S2) to that from the control run (S1) is -33%. These results suggest that the wind farm parameterized by the aerodynamic roughness change resulted in lower concentrations within the wind farm and higher concentrations in the upstream region.

The vertical cross section of hourly $NO_2$ concentrations, simulated by the second model scenario run,

from 1900 to 2200 UTC, November 19 along the transect in the fine domain d03 (Fig. 2) is shown in Fig. 7. Although the maximum concentrations simulated by the S2 run were lower than that from the control (S1) run, particularly within the wind farm, the plumes from the S2 run expanded to the upwind locations of the YWF. This can be seen from the $NO_2$ vertical cross sections at 2000, 2100, and 2200 UTC on November 19 (Fig. 7b-d) which show plume extension from 0 to 20 km compared to the

modeled $NO_2$ plumes in the control run. This is particular evidence at 2000 and 2100 UTC, agreeing with the horizontal distribution of $NO_2$ near the surface (Fig. 6). Figure 8 shows the differences of modeled cross sections of $NO_2$ concentrations between the first and second model scenario runs (the second model scenario run minus control run). In general, higher $NO_2$ concentration differences simulated from the S2 run can be observed at the upwind and interface of the extended YWF, especially

at 0-9 km locations. Lower $NO_2$ differences were observed within the YWF and its downstream region, manifesting again the influences of the wind farm on the spatial distribution of $NO_2$ concentration. The negative differences became more obvious at 2100 and 2200 UTC, respectively. This is likely resulted from stronger easterly and southeasterly wind after 2000 UTC (Fig. 3) which speeds up the atmospheric transport of $NO_2$ from the upstream region to the wind farm.

Figure 9 shows vertical profiles of $NO_2$ from the surface to the 1000 m height, simulated from the control run (S1) and the second model scenario run (S2) respectively at the wind farm grid (44, 52) and upwind grid (50, 48) which is 5 km away from the YWF marked by white star in Fig. 3a, at 2100 UTC, November 19. Within the YWF (Fig. 9a), the S2 model scenario yielded considerably lower concentration (red dash line) below the wind turbine rotor height (~ 40 m) and higher concentration

from this level to the 300 m height compared to that of the control run (solid blue line). The modeled





NO$_2$ concentrations from the S2 run were lower up to 26% than the NO$_2$ level simulated from the control run. At the upstream site (Fig. 9b), the S2 run simulated higher NO$_2$ concentration almost throughout the atmospheric boundary-layer with the concentration level increasing as high as 34% compared to the result from the control run. These results are in line with the NO$_2$ horizontal
distributions and cross sections obtained from the two model scenario runs.

### 3.3   NO$_2$ in YWF by wind turbine drag force parameterization

To confirm the modeling results from the roughness change parameterization for the wind farm, we
replaced this parameterization scheme by the wind turbine drag force parameterization (Eqs. 2-5). This scheme requests the input of the wind turbine density subject to the layout of wind turbines. We set the distance between wind turbines as 500 m in model scenario 3, and in the subsequent numerical scenario run (S4) this distance was extended to 1000 m.

Figure 10 shows the differences of hourly NO$_2$ concentrations at 0600, 1200, 2000 UTC on
November 10 and 0400 UTC on November 20 at the first eta level between the third model scenario run (S3) and the control run (the third model scenario run minus control run) on the same day. Again the NO$_2$ concentrations within the YWF which were simulated by the wind turbine drag force parameterization scheme were lower than that from the control run (S1). The modeled mean concentration within the YWF by the S3 was about 21% lower than that from the control run at 2000
UTC. Mean concentrations at the upwind locations outside the YWF was 13% higher than that simulated by the control run. Overall the values of the concentration differences between the S3 and S1 model scenarios were smaller than the differences between S2 and S1. Higher concentrations were found in the south and southeast of the YWF, differing somewhat from the result by the second model scenario run, as shown in Fig. 6.
The vertical profiles of modeled NO$_2$ concentrations at the two model grids within and at the upwind site from the third modeled scenario run and control run are illustrated in Fig. 11. Lower concentrations at the wind farm grid extending from the surface to the 75 m height were predicted by the 3rd model scenario run with the strongest decline of 8% compared to the control run (Fig. 11a). Above this height, higher NO$_2$ levels extended up to the 200 m height. At the upwind site, the third model scenario run also
predicted significantly higher NO$_2$ concentration than that by the control run, analogous to the result obtained by using the roughness length parameterization scheme (Fig. 9b). The maximum concentration in the vertical is about 27% higher than that from the control run.

We further adopted a low density layout of wind turbines by increasing the distance between two wind turbines from 500 m to 1000 m (the fourth model scenario, S4) and rerun the WRF-Chem with the
same model setups and configurations. Figure 12 shows NO$_2$ concentration differences between the 4th



model scenario run (S4) and the control run (S1) at 0600, 1200, 2000 UTC on November 19 and 0400 UTC on November 20, respectively. As seen, the spatial pattern of the concentration differences subject to the lower density wind turbine setup (1000 m distance) is almost identical to that from the higher density setup (500 m distance). However, the mean $NO_2$ concentration from the model scenario 4 averaged over a region in the YWF, encircled by the red dashed line, was about 16% lower than that from the control run, showing a weaker influence on the changes in $NO_2$ concentration, as compared to the 21% decrease by the higher wind turbine density setup (500 m spacing) from the 3rd model scenario run. At the upwind region of the YWF encircled by the blue dashed line (Fig. 11), the mean $NO_2$ concentration from the lower wind turbine density run (S4) was the same as that from the higher density turbine setup (S3), the both showing 13% reduction of the mean $NO_2$ concentrations from the control run (S1) compared to the model scenario 3 and 4. This is expected because the wind turbine setup is not applicable in the outside of the wind farm.

The vertical profiles of $NO_2$ concentrations from the lowest model vertical level above the surface to the 1000 m height at the model grid (44, 52) within the YWF and grid (50, 48) at the upwind site of the YWF from model scenario 4 and control run are illustrated in Fig. 13. Comparing to the concentration profiles as shown in Fig. 11, the lower wind turbine setup does not markedly reduce the $NO_2$ concentrations within the YWF. The model scenario 4 predicted merely 4% decline from the control run near the surface (Fig. 13a). This scenario also yielded less significant increase in the $NO_2$ concentration at the upwind site of the YWF than that from the higher density turbine setup run with the maximum concentration increase by 20% from the control run, compared with the 27% increase in the higher turbine density simulation (S3).

## 4   Discussions

In this numerical case study, the Yumen-Guazhou Wind Farm, the world largest wind farm located in the western Hexi Corridor, China, was parameterized by the wind turbine induced roughness change scheme and wind turbine drag force scheme, thereby to assess the potential influences of the wind farm on spatial distribution of $NO_2$ within and around the wind farm. Overall our modeling results by making use of these two parameterization schemes predicted higher $NO_2$ concentrations at the immediate upstream and border regions of the YWF and lowered concentrations within the YWF. As aforementioned, a wind farm acts to increase the aerodynamic roughness lengths through two mechanisms. First, the layout and array of wind turbines throughout the wind farm alter underlying surface characteristics (roughness elements) enhancing the roughness lengths within the wind farm. Second, because wind turbines take out momentum proportional to the wind speed, the mean wind speed will be reduced relative to the ambient wind in the wind farm (Emeis and Frandsen, 1993). From



the well-known logarithmic wind law for neutral conditions in the surface boundary layer (~100 m), the reduction of wind speed implies increasing aerodynamic roughness length (Ma and Daggupaty, 2000). As a result, an internal boundary-layer (IBL) could be developed in which the flow characteristics only depend on the new surface roughness. Outside the IBL the flow is identical to the upwind flow (Garratt,

1994; Frandsen, 2007). Hence, the presence of the IBL leads to a step change in the roughness length in the interface between rough (in the wind farm) and smooth (outside the wind farm) surfaces. Such the IBL is particularly evident in the upwind interface. For an air pollutant coming from the upstream of the wind farm on land, the step change in the roughness from the upstream smooth to the rough surface over the wind farm could result in an "overshooting" of the surface stress in the wind farm (Garratt,

1994), slowing down the concentration transport by wind. This would lead to the accumulation of the air pollutants featured by a step change in the concentration at the "edge" (interface) of the wind farm. For the pollutant out of the wind farm to the downstream region, the roughness changes from rough to smooth surface is expected to cause an "undershooting" of the downstream stress which accelerates the pollutant transport in the downwind edge of the wind farm.

Figure 14 is a schematic view of the IBL and the edge effect on an air pollutant passing through a wind farm induced by the mechanic internal boundary layer. In the figure, $h_i$ is the top of IBL, $h_{ss}$ is a sublayer below $h_i$ in which the wind (momentum) has to be adjusted to accommodate the new underlying surface. When the air flow moves from relatively smooth to a rough surface, the wind speed in the IBL will decrease (Garratt, 1994; Bradley, 1968; Elliot, 1958). This deceleration of wind speed

results in the accumulation of air pollution (overshooting), characterized by increasing air concentration in the immediate upwind of the wind farm.

We developed a simple model in the neutral surface boundary-layer to address the changes in the concentration of an air pollutant induced by the roughness changes in a wind farm, given by

$$\Delta \mathbf{c} = -\frac{F_c}{\kappa u_h (c_{\mathbf{D}eff})^{1/2}} \mathbf{ln}\left(\frac{z}{z_{0\mathbf{c\text{-}eff}}}\right), \tag{6}$$

where $\Delta c = c - c_0$ which is the gradient of air concentration of a pollutant at $z_{0\mathbf{c\text{-}eff}}$ and $z$ height in the wind farm, $F_c$ (μg m$^{-2}$ s$^{-1}$) is a diffusive concentration flux (= $\overline{w'c'} = u_* c_*$) where $u_*$ is the fraction velocity (m s$^{-1}$) and $c_*$ is a turbulent scale for concentration (μg m$^{-3}$). $u_h$ is the wind speed (m s$^{-1}$) at the hub height of the wind farm, $c_{\mathbf{D}eff}$ is an effective drag coefficient by summing the surface drag coefficient within the wind farm and the averaged wind turbine drag coefficient, $\kappa$ is the von Kármán

constant (= 0.4), $z$ is the height (0–100 m), and $z_{0\mathbf{c\text{-}eff}}$ is an effective roughness length (m) for concentration, defined by



$$z_{0c\text{-eff}} = 0.1z \exp\left(-\frac{\kappa}{\{[\frac{\kappa}{(\ln(z_{00}/h_b))}]^2 - c_t\}^{1/2}}\right), \tag{7}$$

where $z_{00}$ is an apparent roughness length, $h_b$ is the hub height, and $c_t$ is the averaged wind turbine drag coefficient. Figure 15 displays the vertical profiles of the concentration gradient in the neutral surface boundary-layer (0–100 m) within and outside the wind farm, respectively. Considerably smaller
concentration gradient can be seen within the wind farm compared to that outside the wind farm, forced by increasing drag force under the rough underlying surface in the wind farm.

An interesting feature in the vertical profiles of the simulated $NO_2$ air concentrations in the presence of the YWF by the two parameterization schemes (Figs. 9a, 11a, and 13a) is the lower $NO_2$ level below the hub height (0–70 m) and the higher level above the hub height compared with $NO_2$ concentration
simulated by the control run (the YWF was not taken into consideration). It has been reported that wind farms could significantly slow down the wind speed at the turbine hub-height level and the turbulence generated by wind turbine rotors create eddies which enhance vertical mixing of momentum and scalars (Baiyda et al., 2004). As a result, there would be a wind speed deficit in the neutral boundary layer. The modeled $NO_2$ concentration profiles in the YWF as shown in Figs. 9a, 11a, and 13a are likely associated
with the vertical mixing of air concentrations. Nevertheless, the magnitude of the air concentration deficit in the neutral boundary layer within the wind farm simulated in this model investigation depends on wind farm parameterization. The roughness change parameterization yielded the largest concentration deficit whereas the turbine drag force parameterization with the low wind turbine density produced a moderate deficit. In the immediate upwind region of the YWF, the two parameterization
schemes all predicted notable higher concentrations in the vertical up to 450–600 m height, manifesting significant "edge effect" and the overshooting signature. We wish to point out that here we only discuss the wind profiles over the wind farm in the neutral boundary layer. The diurnal changes in $NO_2$ concentrations presented in the last section took place in the stratified (non-neutral) atmosphere. However, since the wind profiles in the stable and unstable boundary layer can be treated as a departure
from the neutral condition, our interpretations for the "edge effect" should hold for the non-neutral conditions.

It is worthwhile to note that the identification of the "edge effect" or overshooting in the immediate upwind and the undershooting in the downwind region of the wind farm largely depends on the proper locations of upstream emission sources and downstream wind farm which should be aligned with the
wind direction. Figure S2 displays the wind field from the control run and the differences ($\Delta V$) between the perturbed wind field by the wind farm parameterizations (Fig. S2b-d) and the wind field



from the control run (Fig. S2a) at 2000 UTC November 19 at the 4th model level (~100 m). This vertical level is the nearest level to the hub height (70–93 m). At this level, the wind speed should exhibit largest reduction within the wind farm (Emeis, 2010; Frandsen, 2007; Barrie, 2010). As shown, the background wind field in the model domain simulated by the control run (model scenario 1) generated easterly and southeasterly winds across the fine model domain (d03) with stronger easterly winds in the north, except for those model grids near the south boundary of the domain where westerly wind prevailed. Analogous to the previous findings (Fitch et al., 2012), all three wind farm parameterization schemes yield lower wind speed, as shown by -$\Delta V$ across the YWF, particularly in the roughness change parameterization scheme. Outside the YWF, the wind turbine parameterization yielded very small $\Delta V$ (Fig. S2c, d). The roughness change parameterization also predicted -$\Delta V$ across the YWF but positive $\Delta V$ on the south and north lateral boundaries. This feature has also been simulated by Fitch et al. (2012). Figure S3 illustrates the modeled TKE overlapped with vector winds at 2000 UTC November 19 at the 4th model level (~100 m). All three wind farm parameterization scheme predicted largest TKE in the northwestern YWF (Fig. S3 b-d) as compared to non-wind farm (control run) simulations in which no significantly higher TKE was observed (Fig. S3a), corresponding nicely to the largest wind speed deficit and concentration reduction (Figs. 6 and 9). The result is also in line with the TKE field in a relatively smaller wind farm reported by Fitch et al. (2012).

## 5 Conclusions

Extensive model simulations in a case study were carried out to quantify the influence of the world largest wind farm on the spatial distribution of $NO_2$ within and around this wind farm. In this case study, $NO_2$ was emitted from a large-scale iron and steel industry (JISCO) located 110 km southeast to the Yumen Wind Farm (YWF). Under prevailing easterly and southeasterly winds, $NO_2$ concentrations were conveyed from the JISCO to the YWF. Four model scenarios were set up to examine the differences among the modeled $NO_2$ air concentrations with and without the presence of the YWF. In the four model scenario investigations, we implemented two approaches to parameterize the YWF, the roughness length and wind turbine drag force schemes, into the WRF-Chem model. We then compared the differences of modeled $NO_2$ concentrations and concentration cross-sections and vertical profiles within and the immediate upwind of the YWF. Overall the modeling results showed relatively higher concentration at the immediate upwind region and the upwind border region of the YWF, and lower concentration within and the downwind region of the YMF, suggesting an "edge effect" of the wind farm on air pollutants passing over the wind farm. We manifest that the development of the internal boundary layer due to roughness changes induced by the YWF plays a significant role to this edge effect.



We suggested the fluctuations of air pollution over a wind farm might depend on the source locations and proximity, wind speed and direction, underlying surface characteristics, and wind turbine size and the layout of wind turbines in the wind farm. This modeling study is the first investigation of the effect of a wind farm on air pollutants within and around the wind farm. Given the rapid development of wind energy worldwide, the increasing number of wind farms might potentially influence the atmospheric transport of air pollutants and air quality forecasting. More modeling assessments for the influence of wind farms on air pollution should be carried on to assess such potential influences.

**The Supplement related to this article is available online**

*Acknowledgements.* This work is supported by the National Natural Science Foundation of China (grants 41503089, 41371478, and 41671460).

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



**Figure Captions**

**Figure 1.** Location of Gansu province (Shaded yellow area, Fig. 1a) and wind farms in Jiuquan City
(Fig. 1b). Black cross represents YWF and GWF and black dots stand for Yumen City (40°16' N, 97°02'
E) , Guazhou (40°31' N, 95°42' E), and Jiayuguan City (39°48' N, 98°18' E) where the JISCO is located.
**Figure 2.** Nested model domains, including large domain d01 (upper-left figure), the medium size
domain (d02, marked by a white box) covering Guangzhou and Yuman wind farms, and the fine domain
d03 marked by the red box in upper-left figure and by a white box in lower right figure, covering
Yuman wind farm only. Blue shaded area is Jiuquan City. In the d01 domain, the GWF and YWF are
also indicated. These two wind farms are marked by the black cross. The lower-right figure shows the
enlarged d03 area. The red arrow line indicates the transect along which the concentrations
cross-sections are generated (see Sect. 3).
**Figure 3.** WRF-Chem simulatedhourly $NO_2$ air concentrations (ppmv) and vector winds at the first
model level above the surface (~10 m) at 0600, 1200, 2000 UTC November 19, and 0400 UTC
November 20 in the fine domain (d03) from the control run (model scenario 1, S1). The YWF is
encircled by blackdashed line. Two white stars in Fig. 3a stands for two model grids within the wind
farm (44, 52) and outside the wind farm (50, 48) for subsequent discussions. The magnitude of
reference wind speed at 10 m s$^{-1}$ is shown in the upright inner figure.
**Figure 4.** Vertical cross section of hourly $NO_2$ concentration on the transect across the fine domain (d03)
simulated by the control run (model scenario 1, S1) at 1900, 2000, 2100, and 2200 UTC on November
19. The transect is highlighted by the red arrow line in Fig. 2. Terrain height is shown by brown shading,
and the x-axis indicates the length of the transect (km) across the fine model domain d03 and YWF,
bounded by white dashed line, extends from 5 to 25km.
**Figure 5.** Same as Fig. 3 but for the second model scenario run (roughness change parameterization, S2)
using the roughness length parameterization.
**Figure 6.** Differences of modeled $NO_2$ concentrations (ppmv) between the 2nd model scenario run (S2)
and control run (S1) at 0600, 1200, 2000 UTC on November 19th and 0400 UTC on 20th. The wind
field is the same as that shown in Fig. 3 and YWF is encircled by black dashed line. The differences
were calculated by S2 - S1. The deep blue and red dashed lines encircled relatively higher and lower
values of the concentration differences.
**Figure 7.** Same as Fig. 4 but for the 2nd model scenario run using the roughness change
parameterization scheme. The wind farm is bounded by white dashed line.
**Figure 8**. Cross section of the difference of modeled $NO_2$ air concentrations between the first (S1) and
second (S2) model scenario run (S2 - S1) at 1900, 2000, 2100, and 2200 UTC November 19, 2016,





along a transect across YWF, as shown by the red color arrow line in Fig. 3.

**Figure 9.** Vertical profiles of $NO_2$ concentrations at 2200 UTC November 19, at two model grids at (44, 52) within the YWF (a) and (50, 48) in the upstream of the YWF (b), simulated by the control run (S1, blue solid line) and the 2nd model scenario run accounting for the roughness changes in the wind farm (red dashed line).

**Figure 10.** Same as Fig. 6 but for the concentration differences ΔC between the 3rd model scenario run (S3) and the control run (S1), given by S3 - S1.

**Figure 11.** Vertical profiles of $NO_2$ concentration from the surface to the 1000 m height at 2200 UTC November 19 from the control run (S1, solid blue line) and the third model scenario (S3, red dashed line) at (a) the YWF grid (44, 52), and (b) the upwind grid (50, 48).

**Figure 12.** Same as Fig. 6 but for concentration differences between the 4th model scenario run (S4) and the control run (S1), given by S4 - S1.

**Figure 13.** Vertical profiles of $NO_2$ concentration from the first vertical model level above the surface to the 1000 m height at 2200 UTC November 19 from the control run (S1, solid blue line) and the fourth model scenario (S4, red dashed line) at (a) the YWF grid (44,52), and (b) the upwind grid (50,48).

**Figure 14.** Schematic view of the IBL and an air pollutant passing through a wind farm. The IBL and BL change from smaller roughness length 0.01m to large roughness length 1.51m. The red dash line $h$ indicates the PBL thickness, black solid line $h_i$ indicates the IBL, the green dash line $h_{ss}$ indicates a sublayer, $u$ indicates the wind vector, $\delta_s$ indicates the upward displacement of PBL thickness change.

**Figure 15.** Vertical profile of concentration gradient in the neutral boundary layer. The wind speed at the hub height was set as 4 m s$^{-1}$, the surface roughness length was set as 0.01 m, hub height = 60 m. Concentrations were taken as 100 µg m$^{-3}$ at the 1.5 m height and 80 µg m$^{-3}$ at the 10 m height.





Figure 1

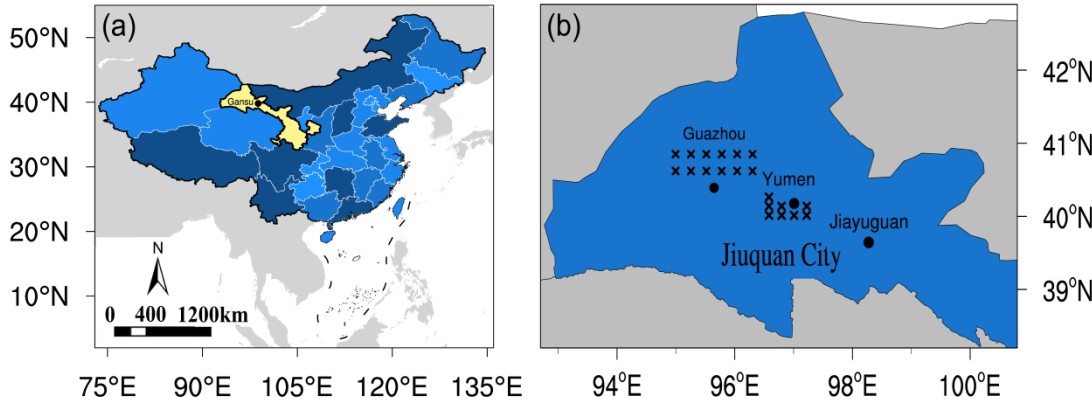

Figure 2

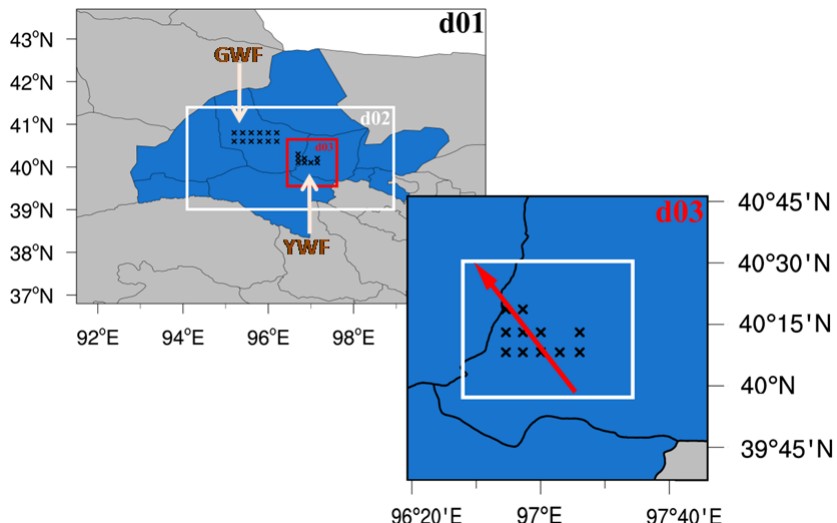



Figure 3

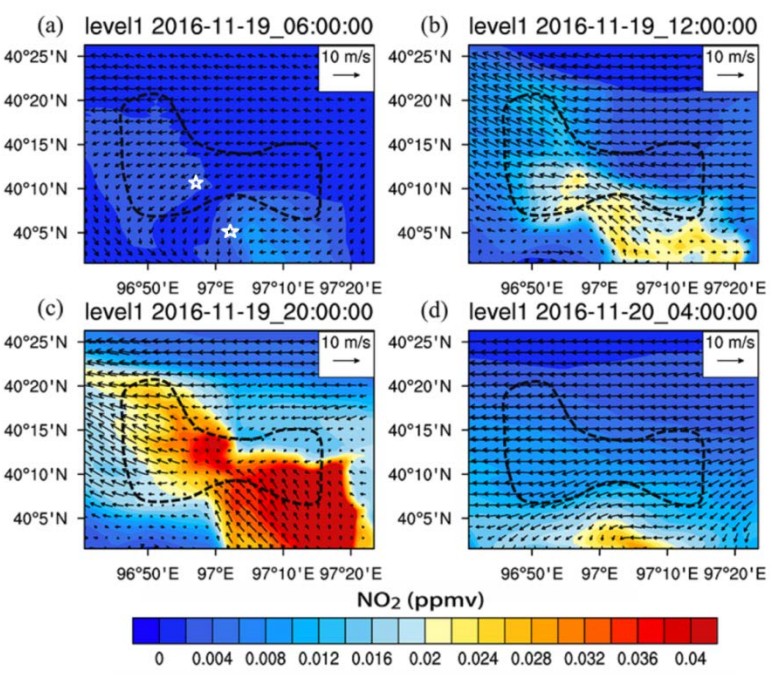

5    Figure 4

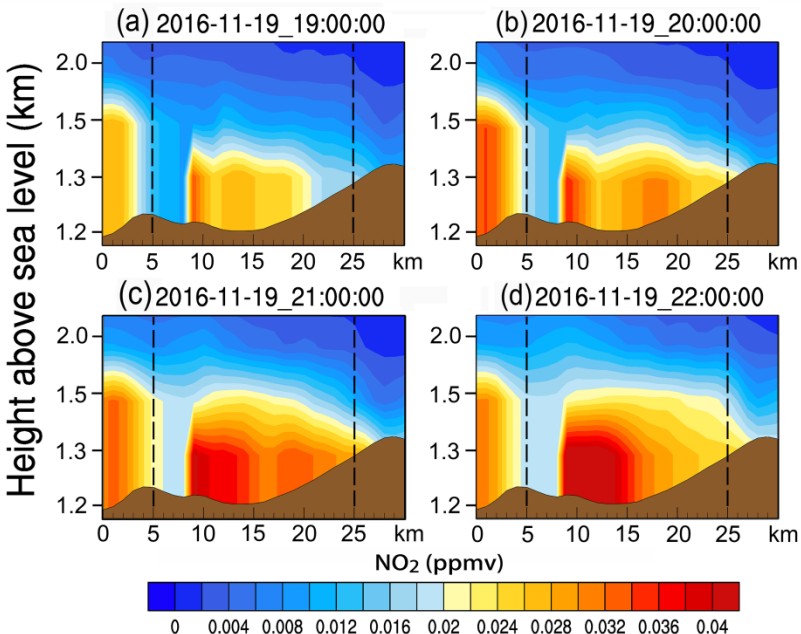





Figure 5

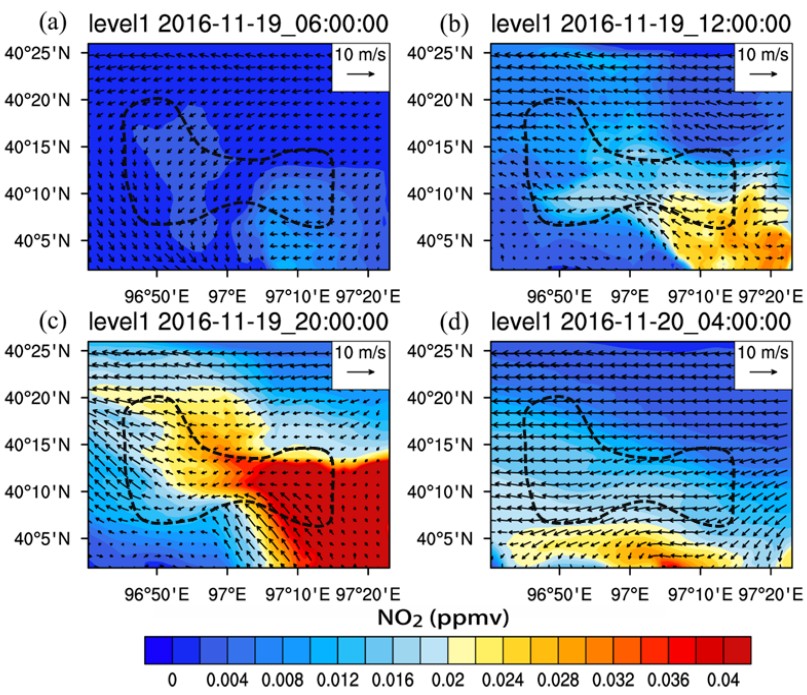

Figure 6

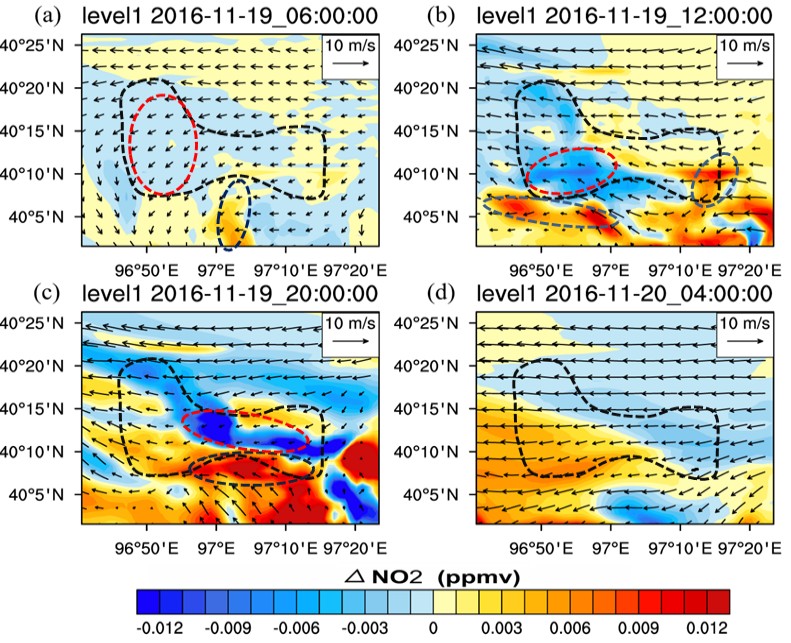





Figure 7

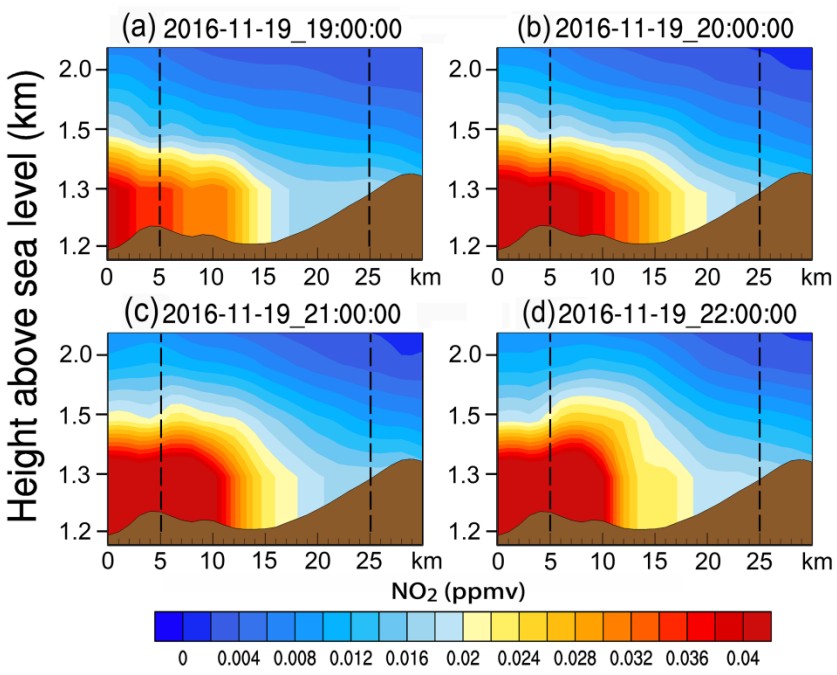

Figure 8

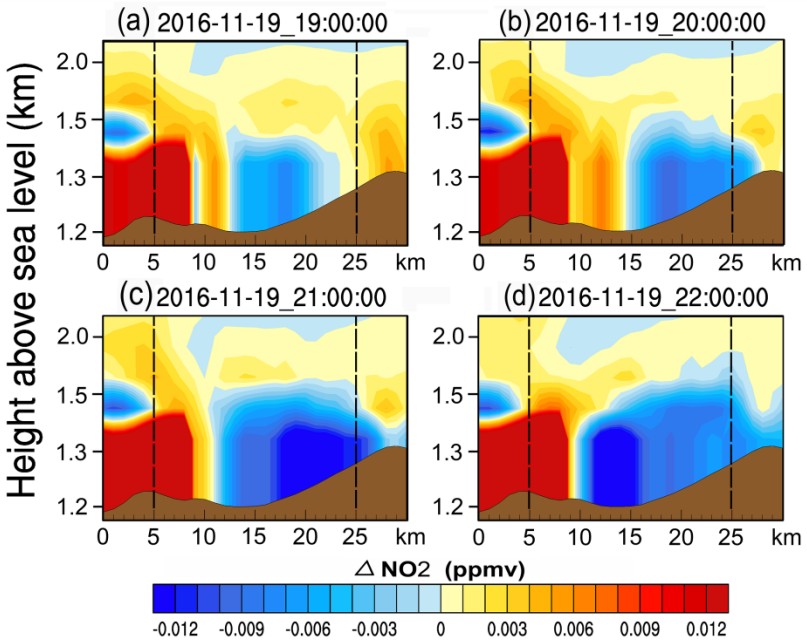



Figure 9

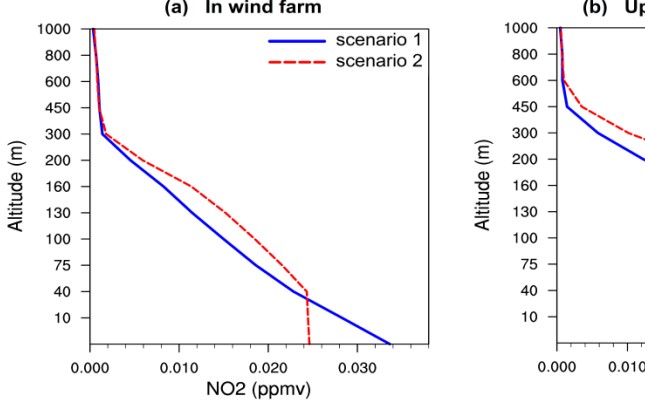

Figure 10

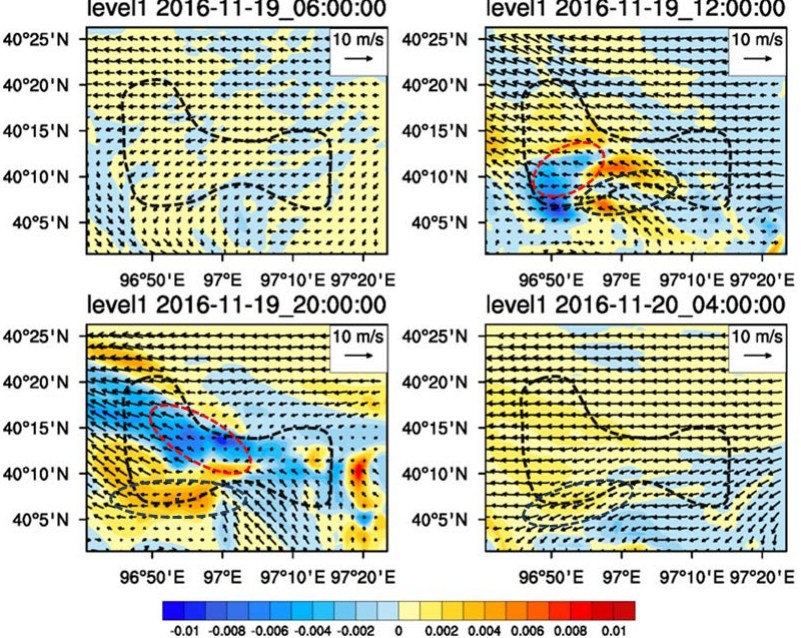



Figure 11

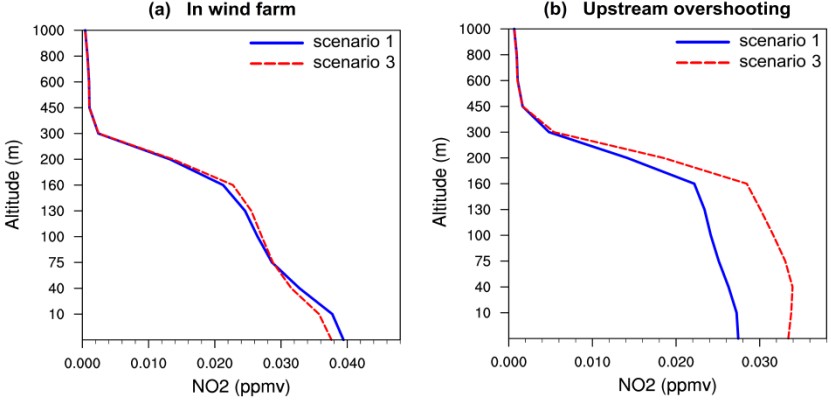

5    Figure 12

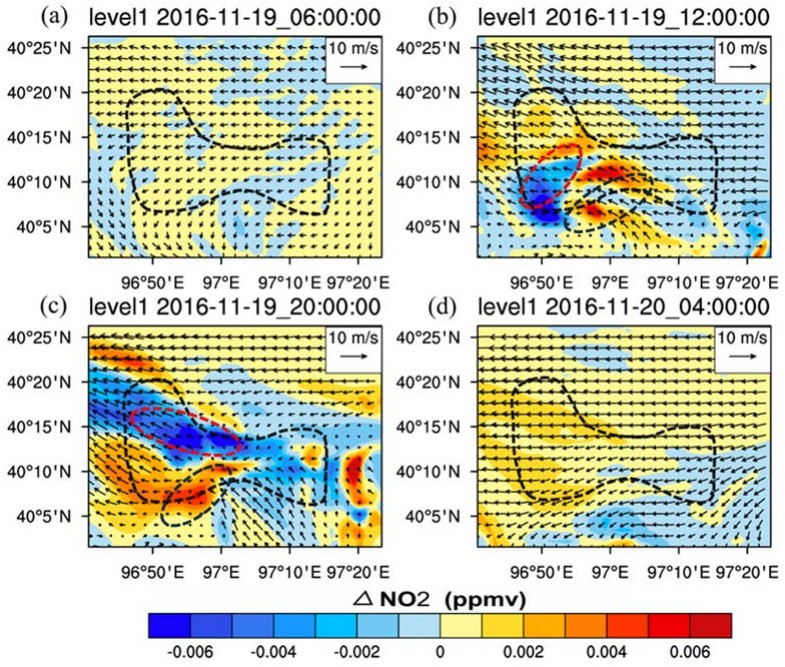



Figure 13

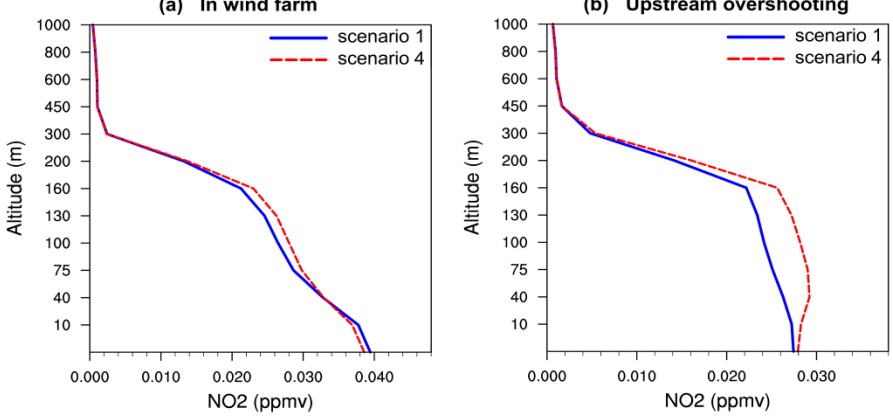

5    Figure 14

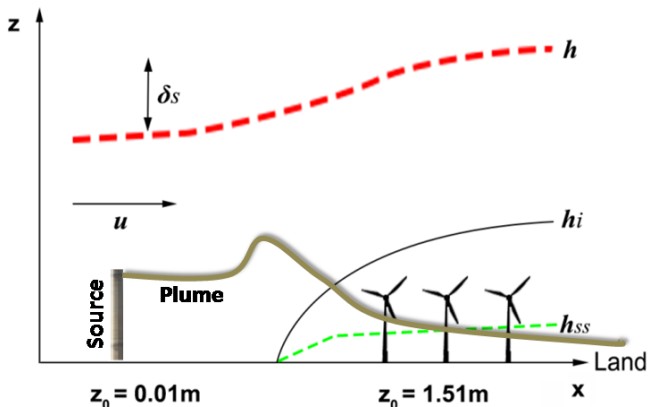



Figure 15

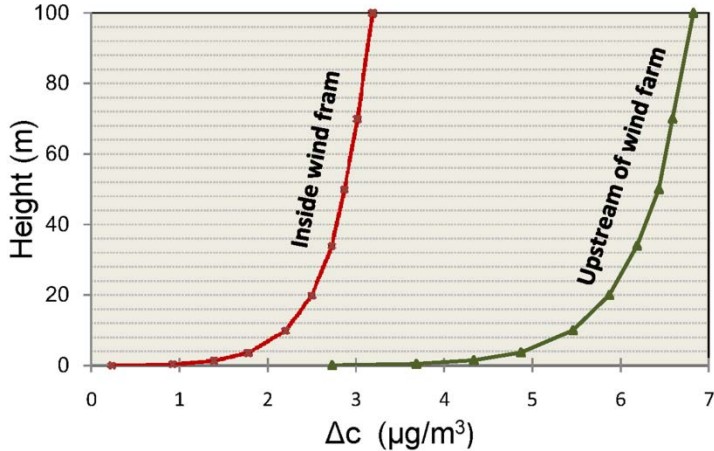

