# Peer review of "Tempo-spatial distribution of nitrogen dioxide within and around a large-scale wind farm – a numerical case study"

_Atmospheric Chemistry and Physics, 2017_

## Referee Comment (RC1) · Anonymous Referee #2 · 9 Oct 2017

This study presents a modeling work investigating the influence of wind farms on the spatial-temporal variation of the air pollutant. The changes in surface roughness length, and the wind turbine density (the layout of wind turbines) over the wind farm, and potential impacts on $NO_2$ concentrations are especially considered. The impacts of wind farm on air pollution have not yet been addressed in most of previous modeling studies, so this is an interesting and scientifically valuable work, which is worthy of publication in ACP. It is clear and well written, with appropriately illustrated. I have a few, generally minor, questions, mostly with the aim to clarify some aspects of the methodology or the limitations associated with the results presented in this study.

[Figure]

General Comments:

1. Section 2.3, page 5, line 23: "..., we modified the geo-data in the WPS and the LANDUSEF table in WRF-Chem model." Could the author clarify and give a bit more details on what variables are modified in the model and the possible uncertainties related?

2. Section 2.4, page 7, the simulated and observed NO2 concentration are compared in Fig. S1, but how is the model performance for reproducing the meteorological fields? The information for meteorological evaluation, especially the wind speed, weed direction, and temperature should be included in the manuscripts. After considering the wind farm parameterization scheme, does the simulated NO2 concentration turn out better or worse when compared the observation?

3. The authors investigate the impacts of the wind farm on the air quality within and around the wind farm regions by a case study and find that the wind farm would lead to the accumulation of the air pollutants featured by a step change in the concentration at the "edge" of the wind farm. But in winter, I think the prevailing wind are mostly westerly wind over these regions, rather than the case in this study, could the authors give some suggestion that how do NO2 levels might change during polluted episode near city regions with the inclusion of the wind farm scheme? How large-scale wind farm may affect the NO2 levels in Jiuquan or Jiayuguan city? Since the city regions have relative high population density, and it's more worthy of concern in big city. The consequences of the changes induced by the large-scale wind farm parameterize on air quality and their implication on human health near large city should be discussed more, at least in the discussion section.

Specific Comments:

1. Organizational suggestion: the simulation runs (S1, S2, S3, S4) are described in section 2.3 currently (page 7, line 1-10), but the simulation case (simulation time, locations) is given in section 2.4. Since all the simulations were performed from November

19th to 24th, 2016, I would suggest put the paragraph (currently page 7, line 1-10) into the section 2.4.

2. In Figure 2, "40°N" should be "40°00'N", "97°E" should be "97°00'E".

3. Page 9, line 17, "control (S1) run" should be "control run(S1)"

4. Page 9, line 22-23, "(the second model scenario run minus control run)" should be "(S2 minus S1)"ãĂĆ

5. Page 10, line 15, "November 10" should be "November 19".

6. Page 10, line 16, "(the third model scenario run minus control run)" should be "(S3 minus S1)". If the simulation runs are named as S1, S2, S3, S4 in the MS, please be consistent throughout the MS.

---

## Referee Comment (RC2) · Anonymous Referee #1 · 10 Oct 2017

Review of "Tempo-spatial distribution of nitrogen dioxide within and around a large-scale wind farm-a numerical case study", by Mo et al.

This paper investigated the tempo-spatial distribution of NO2 concentrations within and around a large-scale wind farm in Gansu, China using WRF-Chem. Adopting two parameterization schemes, the authors found that the wind farm produces an "edge effect", where NO2 are higher in the upwind and border region but lower within the farm and in the downwind region. This paper is well written and structured, and is valuable for evaluation of the impacts of wind farms on atmospheric transport of pollutants and air quality forecasting. I recommend publication in ACP. I have a few minor comments

on the method and result of this study, and enlist them as the follow:

1. Why do the authors set the distance between two wind turbines to 500m and 1000m in the model scenario S3 and S4? Using the real distance between two wind turbines in the Yumen Wind Farm might be more appropriate in the simulation.

2. Why do the authors choose NO2 as the target air pollutant? NOx might be a better target as no chemical evolution is involved. The distribution of NOx could characterize the impact of wind farm on atmospheric transport without the influence of chemical reactions.

3. In the validation part, only the WRF simulation without wind farm parameterization was compared with measurements. The simulations under the two wind farm parameterization schemes should also be validated against measurements to demonstrate that the two schemes could well reproduce the impact of wind farm on the wind field and pollutant distribution in the domain studied.

4. The surface roughness length parameterization treats the wind turbines as pure obstacles while the wind turbine drag force scheme considers the turbines as momentum sink of the wind flow. In reality, the wind turbine could both act as an obstacle and a sink of momentum. Therefore, the effect of wind farm on the pollutant distribution might be a combination of the two schemes to some extent.

---

## Author Comment (AC1) · 24 Oct 2017

Responses to the Referee #1's comments

This paper investigated the tempo-spatial distribution of NO2 concentrations within and around a large-scale wind farm in Gansu, China using WRF-Chem. Adopting two parameterization schemes, the authors found that the wind farm produces an "edge effect", where NO2 are higher in the upwind and border region but lower within the farm and in the downwind region. This paper is well written and structured, and is valuable for evaluation of the impacts of wind farms on atmospheric transport of pollutants and air quality forecasting. I recommend publication in ACP. I have a few minor comments

on the method and result of this study, and enlist them as the follow:

Response: First of all, we would like thank the Referee #1 for his/her constructive comments on our manuscript which helped us to improve our article. Following the comments from the Referee #1, we have revised the manuscript and address all comments from the Referee #1. Our detailed responses and revisions in accordance with the Referee's comments are presented below and in the revised manuscript.

1. Why do the authors set the distance between two wind turbines to 500m and 1000m in the model scenario S3 and S4? Using the real distance between two wind turbines in the Yumen Wind Farm might be more appropriate in the simulation.

Response: The distances between wind turbines in the YWF and GWF are not uniform but range from 300 to 1000 m. The selection of 500 and 1000 m distances aimed to (1) examine the effects of typical layout of wind turbine across the YWF on spatial distribution of $NO_2$ concentrations; and (2) access the response and sensitivity of air concentration to the density of wind turbines in the YWF via two model scenarios. This point has been added to the revised manuscript (page 8, line 1).

2. Why do the authors choose $NO_2$ as the target air pollutant? NOx might be a better target as no chemical evolution is involved. The distribution of NOx could characterize the impact of wind farm on atmospheric transport without the influence of chemical reactions.

Response: We agree with the Referee #1's comment. Since NOx = NO + $NO_2$ and NO can be quickly oxidized to $NO_2$ in the ambient air, NOx is considered to be approximately equal to $NO_2$. In addition, $NO_2$ is on the list of ambient air quality standards and measured routinely at air quality monitoring stations across China. These data can then be used to verify modeled air concentrations. These have been added to the revised paper (page 7, line 20).

3. In the validation part, only the WRF simulation without wind farm parameterization

was compared with measurements. The simulations under the two wind farm parameterization schemes should also be validated against measurements to demonstrate that the two schemes could well reproduce the impact of wind farm on the wind field and pollutant distribution in the domain studied.

Response: Following the Referee's comment, efforts were made to further compare simulated NO2 concentrations from the model scenarios 2-4 with the measured data collected at the Jiuquan air quality monitoring station (Fig. S1). Results are presented in the revised Fig. S1. Overall the modeled and measured data agreed reasonably well but the modeled concentrations from S3 and S4 scenarios illustrate stronger lag behind the measured data. The winds and temperatures were predicted by WRF model. Since WRF model is an operational forecasting model and has been validated extensively, usually we don't need to verify WRF model. Nevertheless, following the Referee's suggestion we compared WRF predicted winds and temperatures with observed data collected at several met stations within the model domain. Results are presented in the revised Supplementary.

4. The surface roughness length parameterization treats the wind turbines as pure obstacles while the wind turbine drag force scheme considers the turbines as momentum sink of the wind flow. In reality, the wind turbine could both act as an obstacle and a sink of momentum. Therefore, the effect of wind farm on the pollutant distribution might be a combination of the two schemes to some extent.

Response: We agree with the Referee's comment. The wind turbine could both act as an obstacle to enhance the surface roughness and a sink of momentum which results in the momentum loss through both surface friction and spinning wind turbine. The two parameterization schemes used in the present study have, to some extent, similar physical background. This point has been added to the revised paper (page 7, line 2).
* * *

---

## Author Comment (AC2) · 24 Oct 2017

Responses to the Referee #2's comments

This study presents a modeling work investigating the influence of wind farms on the spatial-temporal variation of the air pollutant. The changes in surface roughness length, and the wind turbine density (the layout of wind turbines) over the wind farm, and potential impacts on NO2 concentrations are especially considered. The impacts of wind farm on air pollution have not yet been addressed in most of previous modeling studies, so this is an interesting and scientifically valuable work, which is worthy of publication in ACP. It is clear and well written, with appropriately illustrated. I have

a few, generally minor, questions, mostly with the aim to clarify some aspects of the methodology or the limitations associated with the results presented in this study.

Response: We appreciate Anonymous Referee #2 for his or her comments and the constructive criticisms to our manuscript. We have revised the manuscript following the Referee's comments. Our detailed responses to the Referee's comments and corresponding revisions are presented below

General Comments: 1. Section 2.3, page 5, line 23: "..., we modified the geo-data in the WPS and the LANDUSEF table in WRF-Chem model." Could the author clarify and give a bit more details on what variables are modified in the model and the possible uncertainties related?

Response: To highlight the change in the surface roughness length in the YWF (Yumen Wind Farm) by wind turbine layout, we replaced the land use types and surface roughness lengths defined by LU_INDEX and LANDUSEF variables in the geo-data of the WPS by a land use type scheme which takes into account typical land surface characteristics in northwestern China (Zhang et al., 2015) and estimated effective roughness lengths in the wind farm parameterization. Corresponding changes were made in the revised paper (page 5, line 22).

2. Section 2.4, page 7, the simulated and observed NO2 concentration are compared in Fig. S1, but how is the model performance for reproducing the meteorological fields? The information for meteorological evaluation, especially the wind speed, weed direction, and temperature should be included in the manuscripts. After considering the wind farm parameterization scheme, does the simulated NO2 concentration turn out better or worse when compared the observation?

Response: Since WRF model is an operational weather forecasting model and has been validated extensively, we thought we might not need to verify WRF model performance. Nevertheless, following the Referee's suggestion further efforts were made to evaluate WRF predicted meteorological variables (winds and temperature). We

compared WRF simulated winds and temperatures from all 4 model scenarios with observed data collected at several meteorological stations within the model domain. Results are presented in the revised Supplementary. Overall the modeled meteorological fields agree well with the monitored data. We did not find that the modeled winds and temperatures after wind farm parameterization become worth when compared with observations. Since the observed hourly wind direction at local weather stations stands for instantaneous wind direction which is often in equilibrium with underlying surface characteristics and subject to turbulence, it is not straightforward to compared measured wind directions with modeled wind directions which are virtually mean wind directions.

3. The authors investigate the impacts of the wind farm on the air quality within and around the wind farm regions by a case study and find that the wind farm would lead to the accumulation of the air pollutants featured by a step change in the concentration at the "edge" of the wind farm. But in winter, I think the prevailing wind are mostly westerly wind over these regions, rather than the case in this study, could the authors give some suggestion that how do NO2 levels might change during polluted episode near city regions with the inclusion of the wind farm scheme? How large-scale wind farm may affect the NO2 levels in Jiuquan or Jiayuguan city? Since the city regions have relative high population density, and it's more worthy of concern in big city. The consequences of the changes induced by the large-scale wind farm parameterize on air quality and their implication on human health near large city should be discussed more, at least in the discussion section.

Response: The Referee raises an interesting question. Westerly wind prevails in the wintertime across the Hexi Corridor. If a large scale wind farm could disturb atmospheric dispersion of an air pollutant and is located near a city, it may influence the temporal-spatial distribution of the pollutant over the city. This would depend on how far the edge effect could extend in the surrounding region of a large scale wind farm. The edge effect of an internal boundary layer can be estimated via a "fetch-height ratio"

[Figure]

(Garratt, 1994). In micro-meteorology, such the ratio is typically about 1:100 from the rough to smooth surface. In the smooth to rough surface case, the fetch-height ratio is approximately two times greater than that in the rough to smooth case (Garratt, 1994). This suggests that, if the mean obstacle height of the YWF is equivalent to the wind turbine hub height (∼100 m), the fetch over which the edge effect could be extended to would be 10 km. If the westerly wind prevails in winter and knowing that the roughness changes from rough to smooth surface would accelerate the pollutant transport in the downwind edge of the wind farm, we would expect that the eastward transport of air pollutants might influence downwind residential areas, such as Jiuquan and Jiayuguan City in our case (Fig. 1b), located in the downstream of the YWF. However, given that there were no significant emission sources in the upstream of the YWF under the westerly wind regime, the edge effect on the air quality in these two largest cities in the Hexi Corridor was negligible. We have inserted a new paragraph (the last paragraph) in Discussion section to address this question following the Referee's comment (page 14, line 33).

Specific Comments: 1. Organizational suggestion: the simulation runs (S1, S2, S3, S4) are described in section 2.3 currently (page 7, line 1-10), but the simulation case (simulation time, locations) is given in section 2.4. Since all the simulations were performed from November 19th to 24th, 2016, I would suggest put the paragraph (currently page 7, line 1-10) into the section 2.4.

Response: This is a good suggestion! We have moved the last paragraph of section 2.3 to section 2.4 (second paragraph)

2. In Figure 2, "40°N" should be "40°00'N", "97°E" should be "97°00'E".

Response: Figure 2 has been revised.

3. Page 9, line 17, "control (S1) run" should be "control run(S1)"

Response: Corrected. Thanks!

4. Page 9, line 22-23, "(the second model scenario run minus control run)" should be "(S2 minus S1)"ãËŸA ′C

Response: Done!

5. Page 10, line 15, "November 10" should be "November 19".

Response: Corrected. Thanks!

6. Page 10, line 16, "(the third model scenario run minus control run)" should be "(S3 minus S1)". If the simulation runs are named as S1, S2, S3, S4 in the MS, please be consistent throughout the MS.

Response: Done. Thanks!